# An Overview of Golgi Membrane-Associated Degradation (GOMED) and Its Detection Methods

**DOI:** 10.3390/cells12242817

**Published:** 2023-12-11

**Authors:** Hajime Tajima Sakurai, Satoko Arakawa, Hirofumi Yamaguchi, Satoru Torii, Shinya Honda, Shigeomi Shimizu

**Affiliations:** 1Department of Pathological Cell Biology, Medical Research Institute, Tokyo Medical and Dental University, 1-5-45 Yushima, Bunkyo-ku, Tokyo 113-8510, Japan; s924h023@guh.u-hyogo.ac.jp (H.T.S.); arako.pcb@mri.tmd.ac.jp (S.A.); h-yamaguchi.pcb@mri.tmd.ac.jp (H.Y.); toripcb@tmd.ac.jp (S.T.); honda.pcb@mri.tmd.ac.jp (S.H.); 2Department of Biochemistry and Molecular Biology, Graduate School of Science, University of Hyogo, Harima Science Garden City, Himeji 678-1205, Hyogo, Japan

**Keywords:** GOMED, autophagy, ULK1, Golgi

## Abstract

Autophagy is a cellular mechanism that utilizes lysosomes to degrade its own components and is performed using Atg5 and other molecules originating from the endoplasmic reticulum membrane. On the other hand, we identified an alternative type of autophagy, namely, Golgi membrane-associated degradation (GOMED), which also utilizes lysosomes to degrade its own components, but does not use Atg5 originating from the Golgi membranes. The GOMED pathway involves Ulk1, Wipi3, Rab9, and other molecules, and plays crucial roles in a wide range of biological phenomena, such as the regulation of insulin secretion and neuronal maintenance. We here describe the overview of GOMED, methods to detect autophagy and GOMED, and to distinguish GOMED from autophagy.

## 1. Introduction

Macroautophagy (hereafter referred to as autophagy) is a catabolic process in which cellular contents are isolated and degraded using lysosomal enzymes. Canonical autophagy occurs constitutively at basal levels, and is activated by various cellular stressors, including nutrient starvation and growth factor withdrawal, and in response to organelle damage [1,2,3]. Canonical autophagy originates from a portion of the endoplasmic reticulum (ER) membrane [4,5]. On the other hand, we recently identified an alternative type of autophagy that originates from the Golgi membrane [6,7,8,9].

The Golgi apparatus is an organelle involved in the molecular sorting of transmembrane proteins, endo-lysosomal luminal proteins, and secreted proteins. These proteins are synthesized in the ER and undergo protein modifications such as glycosylation during their sequential passage through the *cis-*, medial-, and *trans*-cisterna of the Golgi, where they are prepared to be sent to their appropriate destinations, and are transported by vesicular transport. In addition to these classical functions of the Golgi, we recently discovered a novel intracellular proteolytic mechanism, namely, “alternative autophagy” [6,8]. The morphological features and functions of alternative autophagy are similar to those of canonical autophagy, but it differs from canonical autophagy in terms of the molecules involved, substrates degraded, and biological function, and hence we recently renamed it to Golgi membrane-associated degradation (GOMED) [7,9]. In this review, we introduce the molecular mechanisms and biological roles of GOMED, as well as detection methods of GOMED.

## 2. Discovery of GOMED

Canonical autophagy has been analyzed in detail as a bulk proteolytic mechanism in cells. Canonical autophagy begins with a region of the ER membrane referred to as the omegasome. In the initial stage, an isolation membrane, called the phagophore, is formed. This phagophore membrane elongates and curves to enclose cytoplasmic cargos and organelles, eventually fusing at both ends to form the double membrane structure of autophagosomes. Once autophagosomes are formed, they fuse directly with lysosomes to form autolysosomes, and the enclosed cargos are then degraded by lytic enzymes from the lysosomes [1,2,3] (Figure 1). Because lysosomes contain various types of lytic enzymes, including proteases such as cathepsin, acid lipase, DNase II, and RNase, the cargos are almost completely degraded. It is known that key molecules, such as Atg5 and Atg7, are required for such canonical autophagy to be carried out.

We previously analyzed genotoxic stress-induced canonical autophagy using mouse embryonic fibroblasts (MEFs) from wild-type (WT) and Atg5-deficient (Atg5^KO^) mice [6]. When we treated WT MEFs with rapamycin (which induces autophagy via mTor inhibition), we observed autophagic structures on electron microscopy, as well as protein degradation. On the other hand, no such response was observed in Atg5^KO^ MEFs, confirming that canonical autophagy occurs in an Atg5-dependent manner. However, when Atg5^KO^ MEFs were treated with etoposide (a DNA-damaging reagent), phagophore-like, autophagosome-like, and autolysosome-like structures all appeared as in WT MEFs, and proteolysis was also observed, indicating the presence of an Atg5-independent type of autophagy [6]. According to the definition proposed by de Duve [10], autophagy is the formation of autophagosomes, which are double-membrane structures, and the degradation of their contents by lysosomes, and hence the canonical autophagy-like structures that appeared in the etoposide-treated Atg5^KO^ MEFs, were consistent with the definition of autophagy. Thus, we initially identified this Atg5-independent autophagy as alternative autophagy.

## 3. Comparison of Autophagy and GOMED

A comparison of autophagy and GOMED demonstrated that they share morphological similarities. Both autophagosomes in autophagy and autophagosome-like structures in GOMED are closed compartments surrounded by double membranes. Both autolysosomes and autolysosome-like structures are the compartments that degrade cargos using lysosomal lytic enzymes and are surrounded by single membranes (Figure 1). Another common point is that they are both evolutionarily conserved from yeast to mammals. In terms of differences, however, in addition to their different induction stimuli, the origin of the membrane and the main molecules involved are different. Of particular importance is that the substrate molecules that are degraded are very different, and hence their biological roles are also different. Therefore, GOMED is considered to be a cellular function that is independent from canonical autophagy.

Regarding the inducing stimuli, nutrient starvation and rapamycin treatment, which are commonly used in the analysis of autophagy, strongly induce autophagy, whereas they hardly induce GOMED. On the other hand, stimuli that induce protein loading into the Golgi apparatus, namely Golgi stress, induce GOMED, whereas they hardly induce autophagy. In addition, there are many stimuli that induce both autophagy and GOMED, including those that induce DNA damage [6].

It is widely known that the membrane of autophagic structures originates from the ER [4,5]. However, GOMED originates from *trans*-Golgi membranes. This was demonstrated by electron microscopic analysis, i.e., although there was no change in the morphology of the ER during GOMED, the Golgi apparatus became ministacked and phagophore-like structures were generated from the *trans*-Golgi [6]. Immunoelectron microscopic analysis showed that Golgi-resident molecules are present not only on Golgi membranes but also in GOMED membranes [7]. Recently, it has been shown that Atg9 vesicles formed in the Golgi are required for autophagic initiation [5,11,12]. As GOMED activity was unaffected in Atg9KO MEFs, it might be suggested that the Golgi apparatus has a general key role both in autophagy and GOMED [13]. Functional observations also demonstrated that GOMED is not induced in yeasts, *S. cerevisiae*, when GRH1, a molecule involved in Golgi stack formation, and GRASP65, a molecule involved in folding, are defected [7]. In mammalian cells, for molecules involved in GOMED, such as Ulk1 and Wipi3 to function, these molecules need to be translocated from the cytoplasm to the Golgi [7,14,15].

## 4. Molecular Mechanism of GOMED

The molecular mechanisms of autophagy are well conserved from yeast to mammalian cells, and molecules such as Ulk1 (Atg1 in yeast), Atg9, and Beclin 1 (Atg6 in yeast, a component of the PI3 kinase class III complex) are important in the initial stages [1,2,3] (Figure 1). The subsequent elongation of the phagophore membrane requires WIPI1 or WIPI2 molecules with lipid transfer activity [16,17], and two types of ubiquitin-like binding systems play important roles downstream of these molecules. One is the Atg5 system (involving Atg5, 7, 10, 12, and 16); the Atg5–12 complex is distributed unevenly across the outer membrane of the phagophore membrane, promotes phagophore membrane elongation, and is released from the membrane after the formation of autophagosomes. The other is the LC3 system (involving Atg3, 4, 7, and 8); the LC3-phosphatidylethanolamine complex binds to the phagophore membrane and autophagosome membrane in an Atg5–12 complex-dependent manner [1,2,3] (Figure 1), which contributes to autophagosome formation.

Based on morphological homology, it is likely that several of the above molecules are also involved in GOMED. In fact, knockdown of various autophagy-associated molecules demonstrated that a group of molecules that function relatively upstream in the autophagy mechanism, including Ulk1 and Beclin 1, also play important roles in GOMED [6]. WIPI1 and WIPI2, which function in downstream steps of Ulk1, are not required in GOMED, but their homologues, WIPI3 and WIPI4, are required [15]. These molecules are usually localized in the cytoplasm, but upon stimulation and in a phosphatidylinositol 3-phosphate (PI3P)-dependent manner, they translocate to the Golgi and function in the formation of GOMED structures from the *trans*-Golgi membranes. In clear contrast to the commonality of these molecules, core molecules of autophagy, such as Atg5, Atg7, Atg9, Atg12, Atg16, and LC3, which are involved in autophagosome formation, are not involved in GOMED [6]; instead, molecules such as Rab9 and Dram1 function in GOMED [6,18] (Figure 1).

Regarding Ulk1, which functions in both canonical autophagy and GOMED, it was unclear as to how it regulates these two different cellular functions within a single cell. By performing comprehensive phosphorylation mass spectrometry analysis of the Ulk1 molecule in cells subjected to DNA damage, we successfully found the following facts [14,19,20] (Figure 2). Namely, (1) the 637th serine of Ulk1 is dephosphorylated during both canonical autophagy and GOMED; (2) this dephosphorylation is carried out by the phosphatase PPM1D, which is transcriptionally induced in a p53-dependent manner after DNA damage [20]; (3) during GOMED, the 746th serine of Ulk1 is phosphorylated after the above dephosphorylation reaction; (4) phosphorylation of this serine residue is performed by the kinase RIPK3, which is transcriptionally activated in a p53-dependent manner [14]; and (5) this phosphorylation is not involved in autophagy, because only GOMED is inhibited whereas autophagy is induced by the substitution of this serine to alanine, which prevents phosphorylation (Figure 2). These findings were confirmed by the results that neither proteolytic system is induced in PPM1D-deficient cells [20], and that only GOMED was absent in RIPK3-deficient cells upon genotoxic stress [14]. When the 746th serine is phosphorylated, complex formation between Ulk1, Atg13, and Fip200, which is the initial step of autophagy induction, does not occur. Instead, phosphorylated Ulk1 moves to the Golgi and activates GOMED.

## 5. Degradation of Substrate Proteins by GOMED

The biggest difference between autophagy and GOMED is the difference in their degradation substrates. p62, a well-known substrate of autophagy, is not degraded by GOMED. Conversely, autophagy does not degrade various proteins degraded by GOMED, and hence GOMED and autophagy do not compensate each other’s functions. Because GOMED utilizes Golgi membranes, it is conceivable that the proteins transported through the Golgi are degradation substrates. It can also be predicted that when the Golgi transport pathway is overloaded, GOMED degrades proteins that have become stagnant in the Golgi. A typical example is insulin granules. Insulin is synthesized in β-cells of the pancreas, and are secreted through the Golgi apparatus. Because β-cells sense blood glucose levels and determine the amount of insulin to be secreted, during hypoglycemia, insulin secretion is inhibited, resulting in insulin accumulation in β-cells. In such situations, GOMED is induced to degrade the accumulated insulin granules. In fact, when we replaced the culture medium of Min6 cells, a β-cell line, from high to low glucose medium, insulin secretion was suppressed and insulin granules were degraded by GOMED [7]. The degradation of insulin granules by GOMED was also observed in hypoglycemia-induced mice [7]. Importantly, canonical autophagy is not involved in the degradation of insulin during hypoglycemia.

## 6. Biological Roles of GOMED

There are many reports regarding the biological roles of GOMED, in addition to its role in insulin degradation as described above. For example, GOMED has been reported to protect against intestinal pathogenic bacterial infection, and inhibit the onset of intestinal bowel diseases [21]. Regarding the heart, GOMED improves heart ischemia and obesity-induced heart failure using Rab9 [22,23], which is a molecule that executes GOMED. We also showed that GOMED is required for the elimination of mitochondria from erythroid cells to become mature erythrocytes [24], and for the regulation of insulin secretion in response to blood glucose level [7]. Regarding the brain, we generated mice with brain-specific deletion of the *Wipi3* gene, which is a GOMED-inducing molecule [15], to understand the role of GOMED in the brain. These mice became unable to maintain body posture and developed gait disturbance from about 7 weeks of age (Figure 3A). We analyzed the cerebellum of these mice, which is the brain region that controls locomotion, and found the following abnormalities: (1) degenerative loss of cerebellar Purkinje cells (Figure 3B), (2) abnormal Golgi morphology in neurons before their degeneration, and (3) failure of GOMED induction in cerebellar neurons. Further detailed analyses demonstrated (4) the accumulation of iron and ceruloplasmin in the neurons (Figure 3C). Ceruloplasmin is a copper transport protein, but it also metabolizes iron. In neurons, ceruloplasmin is transported to the plasma membrane via the Golgi apparatus, and functions to convert divalent iron into trivalent iron [25]. Trivalent iron is then excreted from the cell. Because ceruloplasmin is a substrate for GOMED degradation, it is possible that in *Wipi3*-deficient mice, ceruloplasmin is not degraded, and accumulates ectopically in the cytoplasm owing to GOMED dysfunction, resulting in excess trivalent iron and iron deposition [26]. This is thought to be the cause of cerebellar degenerative diseases resulting from iron deposition. It should be noted that static encephalopathy of childhood with neurodegeneration in adulthood (SENDA), a human neurodegenerative disorder caused by iron deposition in the brain, which presents with nonprogressive intellectual disability in early childhood and rapidly progressive extrapyramidal symptoms and dementia in adulthood, is known to be caused by genetic mutations in *Wipi4* (homologous gene of *Wipi3*) [27]. However, *Wipi4*-deficient mice show no iron deposition in the brain and mild symptoms, whereas *Wipi3*-deficient mice show iron deposition in the brain and severe symptoms, and may be a model mouse for SENDA.

## 7. How Can GOMED Be Detected?

Autophagy was first discovered by electron microscopy, but clarification of its mechanism has been facilitated by analysis of the dynamics of the molecules involved, such as LC3, Atg5, and p62. In this section, we describe the methods to detect autophagy and GOMED, and to distinguish GOMED from autophagy.

### 7.1. Electron Microscopic Analysis

As mentioned above, we first identified GOMED from the discovery of autophagy-like structures in etoposide-treated Atg5^KO^ cells [6], indicating that autophagy-like structures can be easily identified by electron microscopy. However, it is difficult to distinguish GOMED structures from autophagy structures. One clue is the structure of the ER and Golgi apparatus, because their morphology is slightly different. In GOMED, many Golgi ribbon structures are ministacked and become distinct within the cell, whereas ER structures are almost normal. In cells lacking Atg5 or Atg7, which are molecules essential for autophagy, autophagy does not occur, so GOMED can be easily detected.

In both autophagy and GOMED, it is important to identify autophagosome(-like) structures to properly assess their dynamics. In the case of ordinary transmission electron microscopy (TEM), it is somewhat difficult to distinguish autophagosome(-like) structures (closed structures) from closing phagophores(-like) structures (open structures). The method we use to differentiate between the two is to analyze the difference in electron density between the interior and exterior of these structures, because the density is different in closed structures but the same in open structures [28,29]. Recently, three-dimensional scanning electron microscopy has been used for the analysis of autophagosome(-like) structures, and this method makes distinguishing open structures from closed structures easier than with TEM [30].

The correlative light and electron microscopy (CLEM) method is an analytical method that superimposes fluorescence images and electron microscope images of the same region to identify the structures indicated by the fluorescence images [6]. To distinguish between autophagy and GOMED, the CLEM method is useful when used with cells expressing mCherry-LC3, because mCherry-LC3 localizes only to autophagy structures and not GOMED structures [6,7,8,9]. If structures observed on electron microscopy colocalize with mCherry-LC3, they are autophagy structures, and if they do not colocalize with mCherry-LC3, they can be considered to be GOMED structures. Unlike mCherry-LC3, EGFP-LC3 is not suitable for this analysis because its fluorescence is attenuated in acidic environments, and it is easily degraded by lysosomal enzymes [31].

### 7.2. Analysis Using Lamp1 (or Lamp2)

Lamp1 is a protein that localizes to the lysosomal membrane. Lysosomes are small structures of 100 nm to 500 nm in diameter, and subsequently become larger autolysosome(-like) structures of 300 nm to 1 µm in diameter. Therefore, autolysosome(-like) structures can be distinguished from lysosomes by the puncta size of Lamp1-GFP fluorescence, or by immunostaining of Lamp1/Lamp2 [6,7]. Because the membrane is visualized by these methods, fluorescence signals are often observed in ring shapes by confocal microscopy. These findings are confirmed by the analysis of the Lamp1-GFP signals using the CLEM method, in which almost all large Lamp1-GFP puncta were localized to autolysosome(-like) structures [6,7]. Note that the GFP portion of Lamp1-GFP (GFP-tagged in C-terminal of Lamp1) is oriented toward the outer rather than the luminal side of autolysosomes, so the effects of acidic environments and lysosomal enzymes do not need to be considered [32]. As with electron microscopy analysis, GOMED can be distinguished from autophagy by the combinational use of Lamp1-GFP and mCherry-LC3, or by using Atg5^KO^ cells [6,14,26].

### 7.3. Analysis Using Special Tandem Fluorescent Proteins

Fluorescent proteins have been developed for the identification of autophagy. When the tandem mRFP-EGFP-LC3 protein is expressed in cells, it is widely expressed in the cytoplasm, and localizes to the autophagic membrane upon autophagy induction. When incorporated into autolysosomes, EGFP is quenched and degraded by the acidic environment and lysosomal enzymes, whereas mRFP is resistant to this, enabling autolysosomes to be identified as a red puncta [33]. EGFP-LC3-mRFP-LC3ΔG is a synthetic protein that is a further improvement of EGFP-LC3, and can be used to quantify autophagic dynamics based on the ratio of EGFP-LC3, which is degraded by autophagy, to mRFP-LC3ΔG, which localizes to the cytoplasm regardless of autophagy [34]. On the other hand, although a method to specifically quantify and evaluate GOMED alone has not been developed to date, it is possible to identify both autophagy and GOMED. When tandem mRFP-EGFP (lacking LC3) is expressed in cells, it is widely expressed throughout the cytoplasm. By the induction of autophagy or GOMED, the EGFP is quenched and degraded in autolysosome(-like) structures, and hence they can be identified as red puncta (Figure 4). In fact, the red puncta of mRFP-EGFP have been shown to colocalize with autolysosome(-like) structures by the CLEM method. [7] In addition, the use of Atg5^KO^ cells enables the evaluation of GOMED alone.

### 7.4. Development of the FLIP-Based Autophagy-Detecting Technique (FLAD)

The series of kinetics in which intracellular substrate molecules are “sequestered” by autophagosomes and “degraded” in autolysosomes is called flux, and its analysis has become essential in recent years [34,35]. We recently focused on the membrane dynamics of “sequestration” and developed a method to measure the dynamics of the generation of autophagosome(-like) structures [36]. Nonselective enclosure of cytoplasmic molecules occurs when phagophore(-like) open structures form autophagosome(-like) closed structure (Figure 5). When fluorescent proteins, such as EGFP and mRFP, are expressed in the cytoplasm and photobleached in a portion of the cell, the fluorescence in this region is quenched. Because these cytosolic fluorescent proteins diffuse more freely than in closed structures, they migrate from the nonbleached region to the photo-bleached region, and the fluorescence of the nonbleached region fades accordingly. If there is a closed autophagosome-like structure in the nonbleached region, it looks like a punctum with strong fluorescence without fading (Figure 5). Therefore, quantification of these puncta makes it possible to measure autophagy and GOMED after sequestration. Because the size of the fluorescent proteins is less than 10 nm, the result that these fluorescent proteins are sequestered without leaking out proves that the sequestration membrane is properly closed [37]. We named this method FLIP-based autophagy detection (FLAD) [36]. When EGFP was expressed in autophagy-deficient cells and GOMED was induced, FLAD visualized EGFP sequestration in autophagosome(-like) structures, and also visualized EGFP quenching and degradation in autolysosome(-like) structures. Because the FLAD method can visualize both autophagy and GOMED, it is also possible to differentiate autophagy and GOMED in a single cell by the combinational use of LC3.

### 7.5. Small Fluorescent Probes

Most of the above-mentioned methods for autophagy and GOMED analysis use fluorescent proteins, which require gene transfer into cells. Therefore, we developed three fluorescent probes, namely DAPGreen, DAPRed, and DALGreen, which accumulate in autophagy and GOMED, in collaboration with Dojindo Laboratory [16,38]. These probes specifically recognize membranes of autophagy and GOMED because they are taken up by membranes and emit fluorescence in a hydrophobic environment, and because they recognize the thickness of the membrane and the PI3P that is taken up. We investigated in detail which stage of autophagy or GOMED these fluorescent probes are capable of labeling, and found the following [16]. DAPGreen labels all structures from the initial stage of autophagy and GOMED with green fluorescence, whereas DAPRed labels structures from the middle stage of autophagy and GOMED (slightly before autophagosome generation) with red fluorescence. Therefore, when DAPGreen and DAPRed are used together, structures in the early stages of autophagy and GOMED can be detected as a single green color, and structures in the later stages of autophagy and GOMED can be detected as a mixture of red and green colors (Figure 6). Furthermore, DALGreen labels only autolysosome(-like) structures with green fluorescence, and when used in combination with DAPRed, the middle-stage structures can be detected in red, and the late-stage structures can be detected as a mixture of red and green colors. Thus, these three types of fluorescent probes can be used to analyze autophagy and GOMED dynamics. When used in combination with LC3, autophagy and GOMED can be differentiated.

### 7.6. Analysis Using Signaling Molecules

In all of the methods described above, it is difficult to differentiate between autophagy and GOMED. Therefore, to analyze only GOMED, it is necessary to use Atg5^KO^ cells or to use LC3 in combination. On the other hand, one method to directly evaluate GOMED is to analyze the dynamics of signaling molecules. Specifically, the phosphorylation of Ulk1 can be used. As mentioned above, during GOMED induction, the 746th serine of mouse Ulk1 (747th serine in humans) is phosphorylated and translocated to the Golgi apparatus [14]. This reaction can be identified by immunostaining with an antibody that specifically recognizes phosphorylated Ulk1^S746^ (human Ulk1 phosphorylation can also be detected) [14]. As this phosphorylation is not involved in autophagy, GOMED can be specifically recognized [19,39].

## 8. Conclusions

In this paper, we reviewed the history of GOMED, its molecular mechanism, and physiological functions, and methods to detect GOMED that have been established to date. However, the entire picture of GOMED is not yet clear. Specifically, there are many issues to be clarified in the future. First, details of the mechanism of GOMED induction. Since several *trans*-Golgi/TGN resident proteins are colocalized on GOMED, there are two main possibilities: First, isolation membrane-like structures of GOMED could be extended from the membrane of *trans*-Golgi/TGN cisternae. Second, it might require the dissociation of *trans*-Golgi/TGN cisterna from the other stacks in GOMED induction. Next, further identification of GOMED substrates. The cargo proteins in Golgi apparatus seem to be the main substrates of GOMED, not by autophagy. It might function as the Golgi quality control pathway, such as misfolded proteins, and be linked to their degradation by GOMED [40,41]. Finally, we are concerned with the involvement of GOMED in various diseases. We believe that the recent methods presented in this review have the potential to unveil these issues. Furthermore, we hope that these advances will lead to the identification of GOMED-specific markers such as LC3. 

## Figures and Tables

**Figure 1 cells-12-02817-f001:**
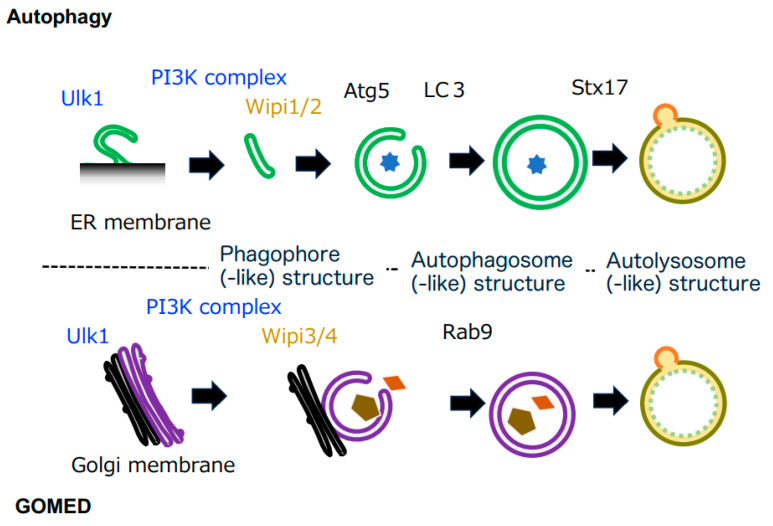
Comparison of autophagy and Golgi membrane-associated degradation (GOMED). Both autophagy and GOMED proceed in the following order: (1) phagophore membrane formation, (2) membrane elongation, (3) autophagosome(-like) structure formation, and (4) autolysosome(-like) structure formation (fusion with lysosomes). Both pathways start with Ulk1, and are induced via the PI3K complex. Autophagy originates from the ER membrane, and the phagophore membrane is elongated via Wipi1/2. Downstream of Wipi1/2, Atg5 is an essential molecule, and LC3 binds to the phagophore membrane in an Atg5-dependent manner, contributing to autophagosome formation. In GOMED, autophagosome(-like) structure formation proceeds in a Wipi3- and Rab9-dependent manner using the Golgi membrane.

**Figure 2 cells-12-02817-f002:**
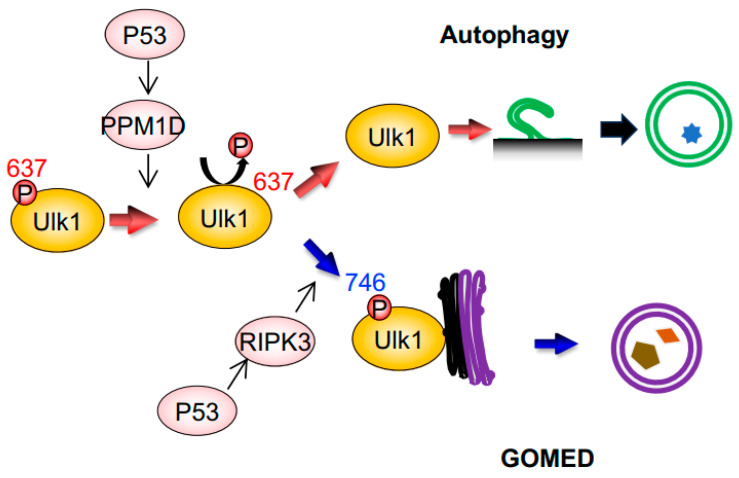
GOMED induction during DNA damage is mediated by the phosphorylation of Ulk1 serine^746^ by RIPK3. Schematic diagram of GOMED induction by DNA damage. DNA damage induces the p53-dependent upregulation of PPM1D and RIPK3; PPM1D induces dephosphorylation of the 637th serine residue of Ulk1, which results in the induction of both autophagy and GOMED. Subsequent phosphorylation of the 746th serine residue of Ulk1 activates GOMED, whereas the absence of this phosphorylation leads to canonical autophagy.

**Figure 3 cells-12-02817-f003:**
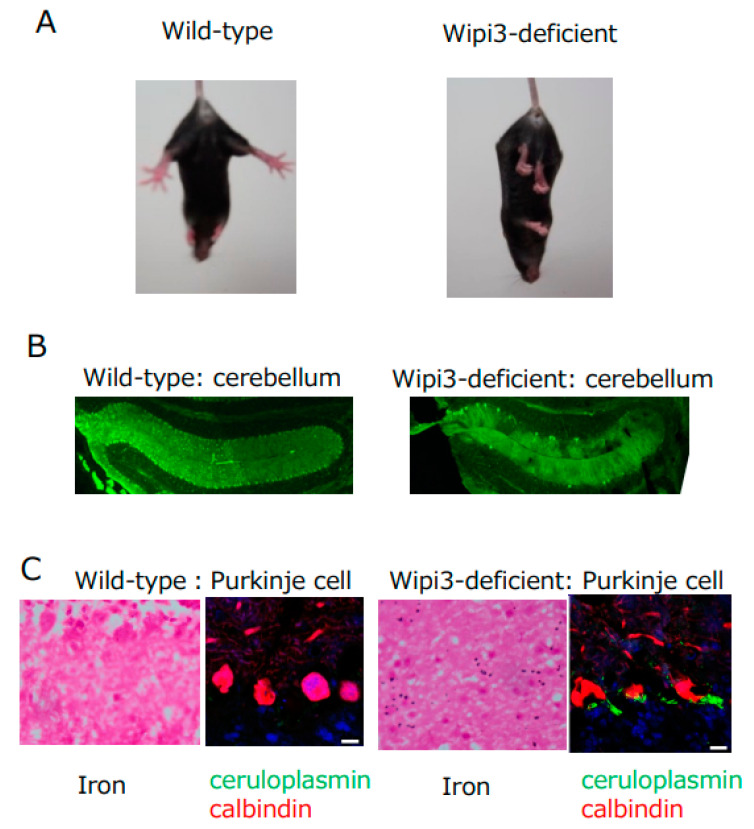
Phenotype of Wipi3-deficient mice. (**A**) Mice lacking Wipi3 show loss of limb-clasping reflex. (**B**) Mice lacking Wipi3 had degenerative loss of cerebellar Purkinje cells (calbindin staining). (**C**) Purkinje cells in mice lacking Wipi3 showed iron deposition and ceruloplasmin accumulation. (Modified from [26]).

**Figure 4 cells-12-02817-f004:**
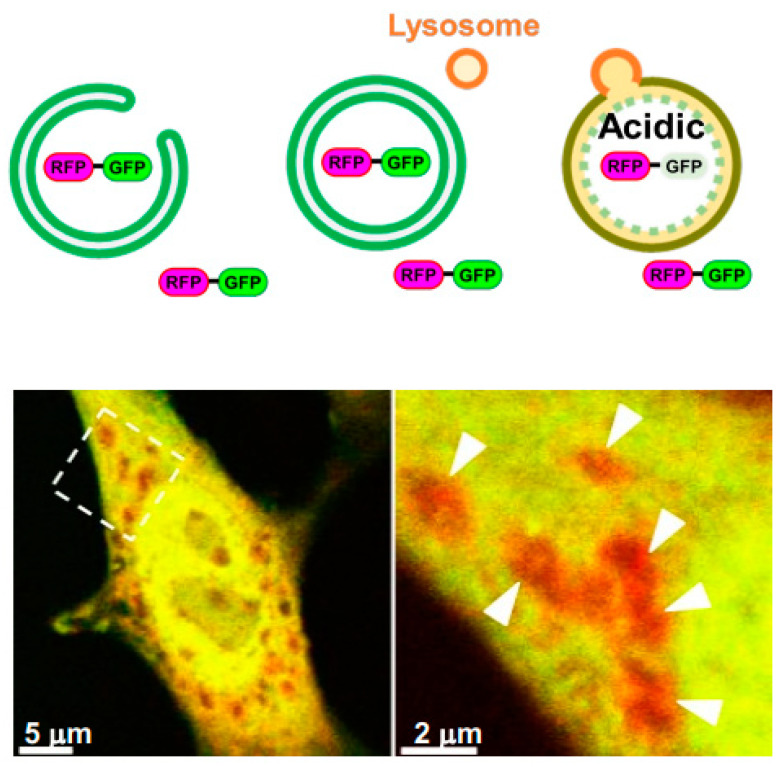
Visualization of autolysosome(-like) structures by mRFP-EGFP. (**top**) Schematic diagram of the method using mRFP-EGFP. mRFP-EGFP can localize on phagophore(-like) structures and autophagosome(-like) structures. In autolysosome(-like) structures, only EGFP is quenched and degraded in an acidic environment. (**bottom**) GOMED is induced in mRFP-GFP-expressing Atg5^KO^ MEFs. mRFP-EGFP can be detected as red only in autolyssome(-like) structures, because only EGFP is quenched and degraded. The right panel is an enlarged image of the boxed area in the left panel. White arrowheads indicate GOMED structures. (Modified from [7]).

**Figure 5 cells-12-02817-f005:**
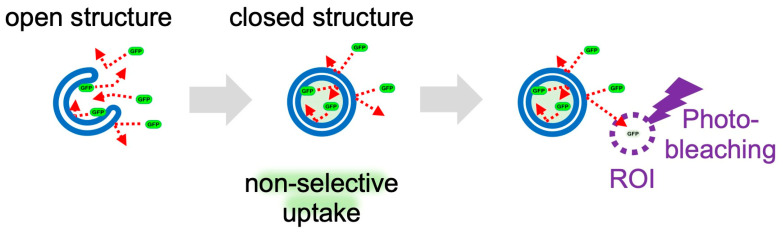
Detection of autophagy and GOMED dynamics by FLAD. Schematic model of FLAD. When GOMED is induced after the expression of EGFP in the cytoplasm, GFP is not confined within the phagophore(-like) structures, but once autophagosome(-like) structures are formed, EGFP is confined and sequestered from the surroundings. When a portion of the cell (ROI) is photobleached with a 488 nm laser, EGFP in the ROI is quenched. By continuous photobleaching, EGFP fluorescence outside the ROI is also reduced owing to free diffusion of the EGFP molecules. On the other hand, EGFP sequestered in autophagosome(-like) structures cannot diffuse, and is detected as puncta that retain their fluorescence intensity.

**Figure 6 cells-12-02817-f006:**
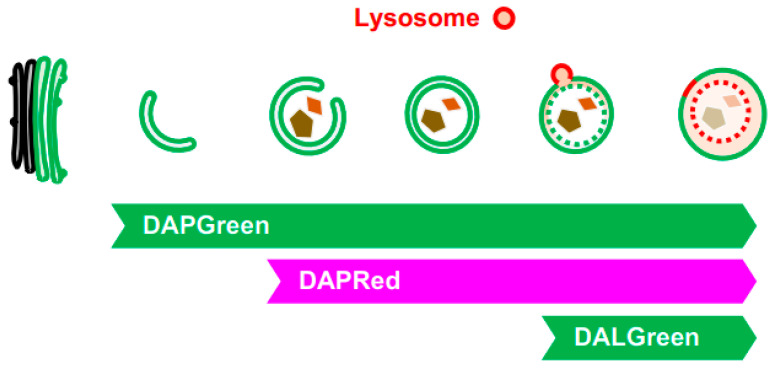
Visualization of GOMED-associated structures by DAPGreen, DAPRed, and DALGreen. Illustration showing the steps of GOMED that are visualized by each probe (bottom left images). When Atg5^KO^ MEFs were stained with DAPGreen/DAPRed and GOMED was induced, early GOMED structures were labeled with DAPGreen alone, and later structures were costained with DAPGreen/DAPRed.

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
