# Peer review of "An Overview of Golgi Membrane-Associated Degradation (GOMED) and Its Detection Methods"

_cells, 2023, doi:10.3390/cells12242817_

Round 1

Reviewer 1 Report (New Reviewer)

Comments and Suggestions for Authors

The article by Sakurai et al. summarises some of the knowledge about autophagic processes in cells, focusing mainly on structures derived from Golgi membranes as opposed to organelles derived from the endoplasmic reticulum (ER). The authors also describe methods for identifying Golgi membrane-associated degradation (GOMED). This is a well written, interesting article for the cell biology community.

A general comment is that the authors mostly refer to their own work and methodologies; it would be relevant to cite and discuss some other reports from different laboratories.

Lines 223 to 238: please include references after each concept explained and differences between cell structures.

Same for paragraph 7-2. (Lines 249-262), 7-4. (Lines 290-312) and 7-6. (Lines 349-359).

References: 11, 21, 23. The lettering shall be adapted as in reference 18.

Author Response

The article by Sakurai et al. summarises some of the knowledge about autophagic processes in cells, focusing mainly on structures derived from Golgi membranes as opposed to organelles derived from the endoplasmic reticulum (ER). The authors also describe methods for identifying Golgi membrane-associated degradation (GOMED). This is a well written, interesting article for the cell biology community.

A general comment is that the authors mostly refer to their own work and methodologies; it would be relevant to cite and discuss some other reports from different laboratories.

# Thank you very much for this high evaluation.

Lines 223 to 238: please include references after each concept explained and differences between cell structures.

# According to this comment, we added new references (ref#28-31).

Same for paragraph 7-2. (Lines 249-262), 7-4. (Lines 290-312) and 7-6. (Lines 349-359).

# We also added new references (ref#32, 35, 37, 39,40).

References: 11, 21, 23. The lettering shall be adapted as in reference 18.

#We adapted the lettering.

Reviewer 2 Report (New Reviewer)

Comments and Suggestions for Authors

This is a very nice review by the authors on the Golgi membrane-associated degradation pathway, GOMED. The identification of this non-canonical autophagy pathway is a recent finding and well worthy of highlighting in a review.  I enjoyed reading the review which was well written and illustrated.  I have some suggestions to further improve the quality and to better integrate the review with the current literature.

1.     The review indicates that classical autophagy originates from ER-derived membranes while GOMED originates from Golgi membranes. Although accurate, it is an over simplification and may give the impression that Golgi membranes are exclusively associated with the GOMED pathway.  Golgi membranes do contribute to the canonical pathway as Golgi-derived vesicles contribute to the expansion and composition of the autophagosome membranes (for example, see review by Tooze, Nature reviews, Mol Cell Biol, 14, 2013). This point should be added to ensure there is no misunderstanding here and that there is a Golgi involvement in both pathways.

2.     There has been a recent identification of a Golgi quality control pathway for misfolded proteins and their degradation (Serebrenik et al, MBoC, 2018, 29, 1284-1298; Hellerschmied et al, MBoC, 2019, 30, 2296).  It would be very useful to include these findings and discuss the relationship between the Golgi quality control degradation pathway and GOMED. Do they differ? If the relationship is unclear at this stage, then this is a useful point to flag.

3.     Further to point 2, the title of the paper implies a overview of “Golgi membrane associated degradation pathways”, however, it is largely limited to the GOMED pathway. The title should be better aligned with the focus of the review.

4.     Does GOMED require dissociated of a Golgi stack (intact?) from the Golgi ribbon or the TGN dissociated from the Golgi? This issue should be further clarified/discussed as I was a little unclear here.

5.     Related to the above, the studies on yeast GOMED and Golgi stack formation (line 105) should be expanded to indicate the yeast species referred to. For example, S cerevisiae does not have defined Golgi stacks whereas S pombe does.  If S cerevisiae, what is the proposed mechanisms for GOMED in the absence of Golgi stacks?

6.     One current challenge as the authors note, is the ability to distinguish between GOMED and autophagy.  In the final section on future goals, it would be worthwhile to suggest future investigations to identify markers or characteristics of the GOMED structures which could provide improvements over current methods to distinguish autophagosomes from the GOMED compartments.

Author Response

This is a very nice review by the authors on the Golgi membrane-associated degradation pathway, GOMED. The identification of this non-canonical autophagy pathway is a recent finding and well worthy of highlighting in a review.  I enjoyed reading the review which was well written and illustrated.  I have some suggestions to further improve the quality and to better integrate the review with the current literature.

# Thank you very much for this high evaluation.

  1. The review indicates that classical autophagy originates from ER-derived membranes while GOMED originates from Golgi membranes. Although accurate, it is an over simplification and may give the impression that Golgi membranes are exclusively associated with the GOMED pathway. Golgi membranes do contribute to the canonical pathway as Golgi-derived vesicles contribute to the expansion and composition of the autophagosome membranes (for example, see review by Tooze, Nature reviews, Mol Cell Biol, 14, 2013). This point should be added to ensure there is no misunderstanding here and that there is a Golgi involvement in both pathways.

# According to this comment, we added the following sentences: “Recently, portion of Golgi essence, Atg9 vesicle, is also required to autophagic initiationref#5, 11, 12. As GOMED activity was unaffected in Atg9KO MEFs, it might suggested that the Golgi apparatus have a general key role both in autophagy and GOMEDref#13. “  (lines110-112)

  1. There has been a recent identification of a Golgi quality control pathway for misfolded proteins and their degradation (Serebrenik et al, MBoC, 2018, 29, 1284-1298; Hellerschmied et al, MBoC, 2019, 30, 2296). It would be very useful to include these findings and discuss the relationship between the Golgi quality control degradation pathway and GOMED. Do they differ? If the relationship is unclear at this stage, then this is a useful point to flag.

# According to this comment, we added the following sentences in the conclusion: “Next, further identification of GOMED substrates. The cargo proteins in Golgi apparatus are seemed to be main substrates of GOMED, not by autophagy. It might be function as Golgi quality control pathway, such as misfolded proteins, and their degradation by GOMEDref#33, 34. Finally, the involvement of GOMED in various diseases.

  1. Further to point 2, the title of the paper implies a overview of “Golgi membrane associated degradation pathways”, however, it is largely limited to the GOMED pathway. The title should be better aligned with the focus of the review.

# Thank you very much for this comment. Because “Golgi membrane-associated degradation”is afull-spell of GOMED, we renamed our manuscript title as “An overview of Golgi membrane-associated degradation (GOMED) and its detection methods

  1. Does GOMED require dissociated of a Golgi stack (intact?) from the Golgi ribbon or the TGN dissociated from the Golgi? This issue should be further clarified/discussed as I was a little unclear here.

# According to this comment, we added the following sentences in the conclusion: “First, details of the mechanism of GOMED induction. Since several trans-Golgi/TGN resident proteins are colocalized on GOMED, there are two main possibilities: First, isolation membrane-like structures of GOMED could be extended from the membrane of trans-Golgi/TGN cisternae. Second, it might require the dissociation of trans-Golgi/TGN cisterna from the other stacks in GOMED induction.“

  1. Related to the above, the studies on yeast GOMED and Golgi stack formation (line 105) should be expanded to indicate the yeast species referred to. For example, S cerevisiae does not have defined Golgi stacks whereas S pombe does. If S cerevisiae, what is the proposed mechanisms for GOMED in the absence of Golgi stacks?

# The reference in the paper focused on S. cerevisiae, not yeast in general. For this reason, the only reference to Yeast in the corresponding section was corrected to S. cerevisiae. (line 113)

  1. One current challenge as the authors note, is the ability to distinguish between GOMED and autophagy. In the final section on future goals, it would be worthwhile to suggest future investigations to identify markers or characteristics of the GOMED structures which could provide improvements over current methods to distinguish autophagosomes from the GOMED compartments.

# According to this comment, we added the following sentences in the conclusion: “Furthermore, we hope that these advances will lead to the identification of GOMED-specific markers such as LC3.”

This manuscript is a resubmission of an earlier submission. The following is a list of the peer review reports and author responses from that submission.

Round 1

Reviewer 1 Report

Comments and Suggestions for Authors

Written by Shigeomi Shimizu who discovered the so-called Golgi membrane-associated degradation pathway (GOMED), this review describes similarities and differences with autophagy and recent findings on this pathway. It is well written and easily understood by the reader.

Minor points:

- line 30: to be sent to

- line 100: what does “the Golgi apparatus is mini-stacked” mean?

- line 118: what is the evidence that GRASP65 is involved in “membrane folding”?

Reviewer 2 Report

Comments and Suggestions for Authors

Review Shimizu et al

Shimizu et al present a review where they refer to their own findings about Golgi-membrane associated degradation (GOMED), a novel pathway that they discovered some years ago. This pathway is very interesting and their findings highly significant, however I think this manuscript is a self-plagiarised version of their review published two years ago in FEBS Journal, doi: 10.1111/febs.16281.

Specifically

-Fig. 1 is exactly the same as Fig. 1 in the FEBS J review and is not referenced properly.

-Order and title of the chapters are the same or very similar

-paragraphs are only slightly changed, but their overall structure is the same. For example, first paragraph in chapter 6 a question is asked, “Then, how does Ulk1 regulate these different cellular functions within a single cell?”. In the FEBS J review in the equivalent chapter, named “role of Ulk1 in GOMED” instead of “Involvement of Ulk1 in GOMED”, the corresponding question at the same position is “Then, how does ULK1 regulate these different autophagy pathways?”

The authors should be familiar with good scientific practice that excludes self-plagiarism. A fully revised version should therefore be clearly different from published ones and has to make clear what its add-on to previous reviews is. Also, I think the review would benefit from some critical assessment of the findings about GOMED. Are there other groups confirming the findings (I didn´t find any), is GOMED widely accepted in the autophagy field, what are the limitations of the studies so far, what needs to be done etc.

I´m sorry to be that negative, but see no other option.

Comments on the Quality of English Language

I have no further comments

Reviewer 3 Report

Comments and Suggestions for Authors

The review presents a comprehensive discussion of "Golgi membrane-associated degradation", a topic to which the author's group has made significant contributions. Given this, the preponderance of references to the group's own work is understandable.

The structure and precision of the review is commendable, with a clear focus on the group's findings. The authors have avoided unnecessary speculation and instead stuck to established facts, which adds to the reliability of the review.

It seems to me that most of the data and figures presented in this review have already been published in other sources. If this is the case, it would be appropriate to attribute the authorship of this review to the individual(s) who compiled and wrote this specific review, rather than to the entire research group.

Reviewer 4 Report

Comments and Suggestions for Authors

See attached file
